# Preserving Fragile History: Assessing the Feasibility of Segmenting Digitized Historical Documents with Modulation Depth Analysis

**Patrick Zippert** *,† , **Felix Binder** † and **Tino Hausotte**

Department of Manufacturing Metrology, Friedrich-Alexander-Universität Erlangen-Nürnberg,
91052 Erlangen, Germany; felix.binder@fmt.fau.de (F.B.); tino.hausotte@fmt.fau.de (T.H.)
* Correspondence: patrick.zippert@fmt.fau.de
† These authors contributed equally to this work.

**Abstract:** Historical documents are often severely damaged, making it impossible to open them manually without causing further damage. To address this challenge, computed tomography (CT) has emerged as a non-destructive method to explore the inside in a different way. However, the use of ionising radiation in CT scanning raises concerns about its impact on fragile historical documents. This study presents a methodology that uses a test object to conduct preliminary investigations to evaluate the capability of a CT scanner for digital preservation of historical documents. By assessing the feasibility and determining the setting parameters in advance, the X-ray exposure to historical documents can be minimised. For this purpose, a large dataset of inter-page distances was obtained from CT scans of a specially developed test object. The results obtained show a consistent correlation between the page-to-page distances and the derived modulation depths. This method offers great potential for assessing the separability of the pages of historical documents even before they are exposed to radiation for digitisation. Overall, this study helps to reduce the impact of X-ray radiation on sensitive historical documents during digitisation using CT, with the aim of preserving this fragile cultural heritage for future generations.

**Keywords:** protection of cultural heritage; preserving historical documents; industrial X-ray computed tomography; feasibility analysis; page segmentation; modulation depth



## 1. Introduction

Historical documents are considered cultural treasures that provide a window into the past. They help us to understand bygone events, ideas, and people that shaped our present world [1]. They are particularly worth protecting so that we as a society and future generations can benefit from their insights.

An example emphasising the importance to preserve historical documents is provided by the Lisbon earthquake in 1755. In this tragic event on 1st November, a massive earthquake destroyed the fourth largest city at that time in Europe. The resulting tremors, fires, and tsunami waves also destroyed the Royal Library of King D. João V, which contained detailed records of Vasco da Gama's voyages and thus evidence of discoveries, trade, and colonisation of the Atlantic by the Portuguese [2]. Another famous example in history is the disappearance of the Library of Alexandria, which left a gap in our knowledge of the past and a loss of cultural identity [3,4].

However, preserving historical documents is not an easy task. Many of those books have become very fragile over the centuries due to natural ageing processes or exposure to external influences. In some cases, it is not even possible to open them without causing further damage. Therefore, new ways have to be found to recover and preserve the content of such fragile and unique books.

One method of making historical documents legible again is to scan them using X-ray computed tomography (CT). CT is a suitable recovery method for text written with metal-based inks that have been widely used in the past, like iron gall ink. Metal-based inks have a higher attenuation coefficient than their carrier material (e.g., cellulose), which results in a good contrast image using CT. Therefore, it is possible to use reconstructed volumes to visualise and digitize their contents with a single scan [5,6]. Seales et al. [7] have demonstrated the advantage of this technology in the study of the Herculaneum papyrus scrolls, which were buried by the eruption of Vesuvius in 79 AD and cannot be unsealed otherwise.

A downside of CT is the use of ionising radiation, which is known to potentially cause damage to the examined materials. However, it is uncertain whether historical documents will be damaged by digitisation using CT [8]. Particularly valuable historical manuscripts are usually stored in libraries, archives, and other cultural institutions where strict environmental conditions prevail. As these institutions are responsible for preserving these sensitive pieces, they are reasonably cautious about considering novel analytical methodologies [9].

To address such concerns, a prior study demonstrates that it is unlikely to cause permanent damage to well-preserved documents in a single CT scan. Nevertheless, an additional ageing effect occurred that visibly yellowed the examined documents if scanned several times [10].

For this reason, a procedure has to be found to minimize the number of scans required for sustainable preservation of historical documents using CT. A key objective towards such a procedure is to design a reproducible test method that enables a pre-evaluation of optimal CT scanning parameters with no exposure to radiation of the target document. Furthermore, a test procedure has to provide a suitability check of the measurement system, including an indicator if the resolution of the setup is sufficient for content recovery. Both objectives aim to significantly reduce the number of necessary scans in order to increase trust in the preserving capability of CT measurements.

The parameters for scanning historical documents depend on a variety of factors, like the size of the books and the used materials and the available CT system. In past centuries, people did not solely write on paper but also on materials such as leather, papyrus, or parchment. Even the paper produced in those days was made from different pulps, such as cotton, hemp, or flax [11,12]. Those different materials and sizes require individual parameter settings, like exposure settings or positioning in the measurement chamber, to achieve good measurement results. The quality of the reconstructed volumes may vary between CT systems due to differences in hardware or software components. Consequently, setup parameters cannot be generalised and applied to all samples of the same type.

Once a historical document has been scanned using CT, processing the volume data is a major challenge. In order to extract the essential written content from the pages of a book, the individual pages must be distinguished from each other [6]. An important factor in determining the separability of individual book pages is the resolution of the scanning system. If the resolution of the volume data is insufficient, the individual pages in the volume data cannot be separated from each other. Consequently, it is necessary to find a suitable separation criterion that enables to assess the separability of scanned historical documents based on the resolution of the volume data. In this way, it is possible to estimate in advance whether the set parameters meet the requirements for later side separation and content extraction.

In this study, we investigated a reproducible test procedure to address this need and allow conclusions to be drawn about the subsequent separability of scanned historical documents under similar scanning conditions.

## 2. Materials

In order to evaluate the scanning parameters and to assess the separability of digitized historical documents using CT, a test object was designed that consists of stacked circular

paper representing book pages (Figure 1). A round shape was chosen to reduce the effects of varying transmission lengths. The paper pages are aligned parallel to the direction of transmission, as is the case with scanned historical documents, to achieve a good contrast between the pages and ink [8].

Paper consists of fibres that form a structured matrix. Due to inherent factors such as manufacturing processes and varying environmental conditions such as humidity or temperature, internal stresses can occur within the paper, resulting in a certain degree of waviness. To ensure the horizontal alignment of the paper sides during the measurement and to create a uniform distance between the papers in the stack, the sample is pressed through an acrylic glass clamp.

The individual paper discs (diameter 31.75 mm) consist of conventional paper (with grammage of 120 g/m$^2$) made from wood pulp with a thickness of approximately 130 µm. In the past, paper was handcrafted using an elaborate process, and the thickness varied greatly depending on the craftsman. For example, the thicknesses of ancient papers from the Corpus Chartarum Italicarum examined in [13], dating from 1293 to the middle of the 20th century, ranged from 140 µm to 250 µm. Our test specimen is in the lower range of these historical document thicknesses as it is assumed that thicker papers can also be separated when the separation limits are reached. To estimate the smallest possible distance between two paper pages within a book, the paper discs of the test sample are separated by different distances. This is achieved by using rings from the same material to create an inner space between the full circles (Figure 1).

The arrangement of paper discs (book pages) and paper rings (spacers) of the test object can be divided into three different patterns: the starting/closing pattern, the variable distance pattern, and the symmetry break pattern (Figure 2).

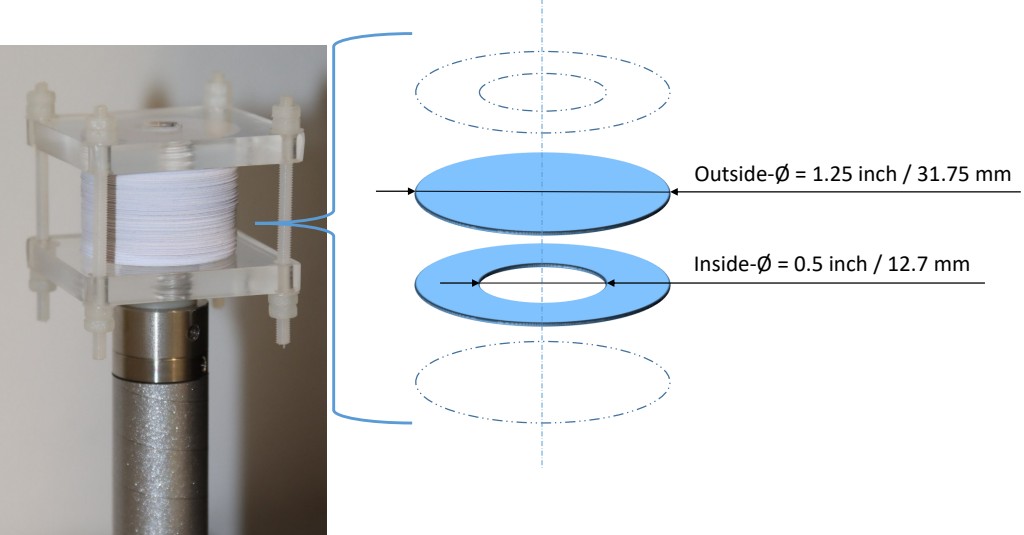

**Figure 1.** Scan-specimen consisting of a paper stack. Paper rings separate the paper pages (full circles) at a defined distance from each other. The grammage of the full paper discs is 120 g/m$^2$ for a thickness of 130 µm; those of the paper rings vary between 120 g/m$^2$ (130 µm), 160 g/m$^2$ (174 µm), and 200 g/m$^2$ (196 µm).

To prevent transition effects from the clamping to the tested paper, 10 circular paper rings with the grammage of 120 g/m$^2$ and an average thickness of 130 µm are positioned at the beginning and at the end of the investigated paper stack section (Figure 2). This configuration also ensures that the variable distance pattern (including the full circles) is clearly recognisable for later evaluation.

The variable distance pattern forms the core of the test object with an arrangement of paper discs (Figure 2, Type A) and a different number of paper rings using varying thicknesses (Figure 2, Type A, B, C). This allows the paper discs, which imitate the pages of

a book, to be placed at different distances from each other. A total of 9 spacing combinations with three different paper thicknesses were created by the paper rings, separating the paper discs at a distance of 130 µm to 392 µm. For a statistical analysis of the spacing, the variable distance pattern was repeated 4 times in the entire paper stack.

To incrementally increase the distance between the full circles (paper discs), the thickness and the number of rings between are adjusted. In particular, paper rings with weights of 120 g/m² (130 µm), 160 g/m² (174 µm), and 200 g/m² (196 µm) are used (Figure 2). In addition, a symmetry break (larger gap between paper discs) consisting of 20 paper rings with a grammage of 120 g/m² is included in the paper stack to avoid symmetrical influences due to the centering of the specimen in the cone beam. Furthermore, the determined gap was used to normalise the generated signal in the subsequent evaluation procedure and provides a visible indicator of the measured orientation within the reconstructed volume.

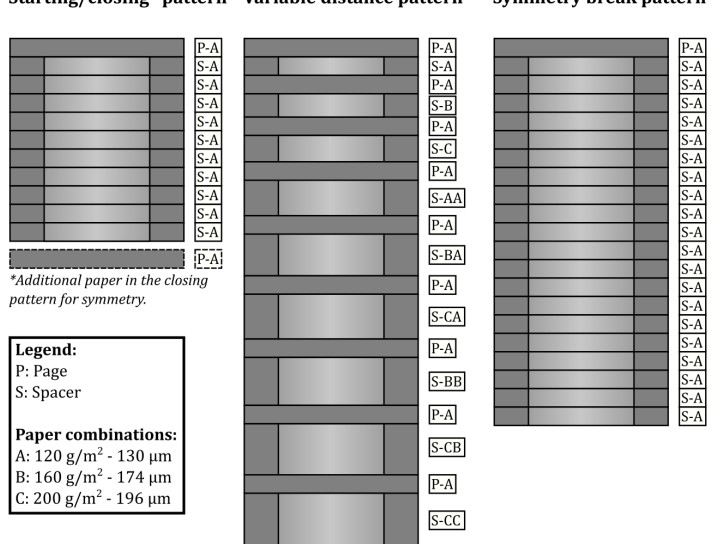

**Figure 2.** Section view schematic of the constructed gap patterns with three paper grammages (thickness 130 µm, 174 µm, and 196 µm). The variable distance pattern is used four times. Between each spacer *S* (consisting of different combinations of paper ring thicknesses) is a 130 µm circular paper *P* (Type A).

## 3. Methods

### 3.1. Measurements Using Industrial Computed Tomography

The paper stack specimen from Section 2 was measured 3 times using an industrial CT system (Zeiss Metrotom 1500). The test object was placed close to the source to achieve a high magnification (see Figure 3). A copper filter was used to attenuate low energy radiation and reduce beam hardening effects. A summary of the measurement parameters employed can be found in Table 1.

The resulting projection stacks were reconstructed using a Feldkamp–Davis–Kress (FDK) algorithm provided by the commercial software CERA (Siemens Healthineers AG, 91052 Erlangen, Germany ). The reconstructed volume has a voxel size of 42.1 µm, which is roughly $\frac{1}{3}$ of the smallest paper thickness (A: 130 µm) and therefore close to the resolution limit.

**Table 1.** Measurement settings for the test specimen.

| Assembly | | Source | | Detector | |
|---|---|---|---|---|---|
| SRD: [1] | 289.77 mm | Voltage: | 170 kV | Size: | $(409.6 \times 409.6)$ mm$^2$ |
| SDD: [2] | 1376.37 mm | Current: | 250 µA | Pixel: | $(2048 \times 2048)$ px |
| Orientation: | Not tilted | Pre-filter: | Cu 0.25 mm | Exposure: | 2000 ms |
| Magnification: | 4.75 | Spot size: | 43 µm | Gain: | 8× |
| | | | | Binning: | $1 \times 1$ |
| | | | | Averaging: | 2× |
| | | | | Projections: | 2050 |

[1] Source to rotation axis distance. [2] Source to detector distance.

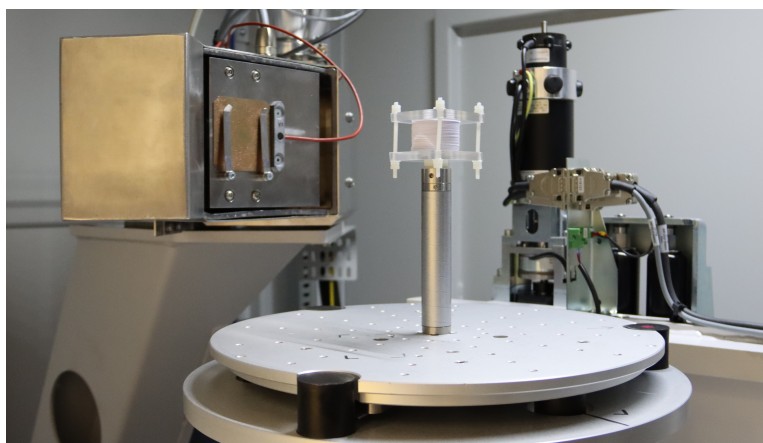

**Figure 3.** Experimental measurement setup: a stack of paper pressed into a clamp out of acryl glass positioned in front of the X-ray source on a rotary stage.

### 3.2. Approach for Separating Arbitrarily Oriented Book Pages Using Modulation Depth Analysis

The research question for this study is: can the distinction between two paper pages be estimated before a CT measurement of an ancient document? Such an a priori assessment is beneficial to reduce the risk of damaging the measurement subject due to radiation or transport. As described in [8], a separability of book pages strongly depends on the resolution of the measurement. In the field of dimensional CT, there are three commonly used values to characterize a resolution regarding [14]. Those are the voxel size, the structural resolution, and the metrological structural resolution (MSR).

The voxel size is the size of the smallest element of a reconstructed volume grid and usually set to match the size of the detector pixel pitch regarding the used magnification of the setup. The structural resolution is mostly used for contrast-based distinctions either within projections or within the reconstructed volume. Contrast differences depend on the attenuation of a given material in CT, which relates to the irradiation pathway and the density of the material. The third category of commonly used resolutions within CT is the MSR, which is focused on a holistic measurement, which includes the application of a surface determination on the reconstructed volume and utilization of a measurement plan.

Each resolution can be used to characterize a CT system for specific cases and is the subject of current research. However, in case of historical documents, none of these resolutions can provide assurance regarding the ability to separate the pages of a scanned book. For instance, voxel size can be used to estimate a good signal strength for the setup but does not provide a good criterion to account for unsharpness. As mentioned by [15], the magnification of a setup influences either the unsharpness contribution of the source or the detector. The unsharpness arising from the X-ray source depends on the shape and intensity distribution of the X-ray focal spot [16]. The unsharpness of the detector is based on its structure and the used active materials [16]. Since the separation algorithm of [8] is directly based on the reconstructed volume data and no surface determination is performed, the MSR cannot be used either. Typical contrast-based structural resolutions,

however, are based on regular structures, which does not apply to irregularly deformed document pages found in genuine historical artefacts.

Therefore, we investigate in this work an approach to separate arbitrarily oriented book pages based on the resulting contrast signal within a reconstructed volume. For the present study, a workflow is described to filter, smooth, and extract contrast signals from a paper stack specimen, which is similar to the task of measuring historical books. The core of this investigation is based on the modulation depth ($M_{DIP}$), defined in the ASTM standard E2002-22 [17], which describes the relation of two adjacent signal peaks ($P_{Left}$, $P_{Right}$) and their interacting valley $V_{Centre}$ due to unsharpness (compare Figure 4).

$$M_{DIP} = 100\% \cdot \frac{P_{Left} + P_{Right} - 2V_{Centre}}{P_{Left} + P_{Right}} = 100\% \cdot \left( 1 - \frac{2V_{Centre}}{P_{Left} + P_{Right}} \right) \tag{1}$$

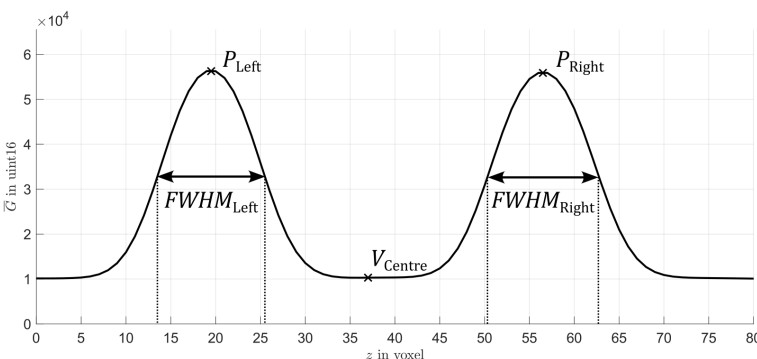

**Figure 4.** Visualization of the modulation depth with two Gaussian-shaped signals.

### 3.3. Analyzing Convergence Behavior of Contrast Signals Using Modulation Depth

Since the pages of historical books are handcrafted and may be weather-stained or otherwise decayed over time, the actual surface of a single page has to be assumed as an arbitrarily shaped closed surface with a variable material density and composition. Therefore, the sides tend to bend even though the paper stack is compressed by the clamping, resulting in randomly aligned adjacent surfaces (see Figure 5).

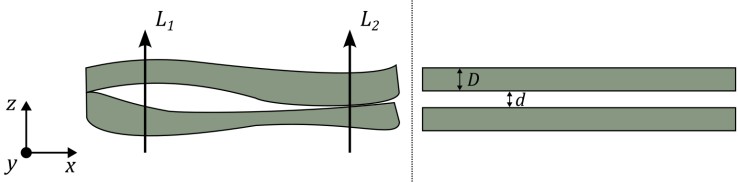

**Figure 5.** Left: schematic slice-view of two historical pages with arbitrary width and surface orientation. The gap between two pages along $L_1$ and $L_2$ impacts the separability. Right: idealized representation of two pages with a known thickness $D$ and a constant gap $d$.

The attenuated intensity is converted in CT with a detector array into a contrast signal, and, in the case of the measurement setup described in Section 3.1, into a 16-bit unsigned integer (uint16) grey value $G$.

As simulated in Figure 6, the mean grey value $\overline{G}$ along $z$ does not follow a simple convolution of two Gaussian signals since the resulting valley at distances $d$ smaller than the full width at half maximum (FWHM) of the Gaussian function are smaller than the actual paper peaks (compare Figure 6). Therefore, $M_{DIP}$ is applicable even for close page distances in case of a sufficient resolution.

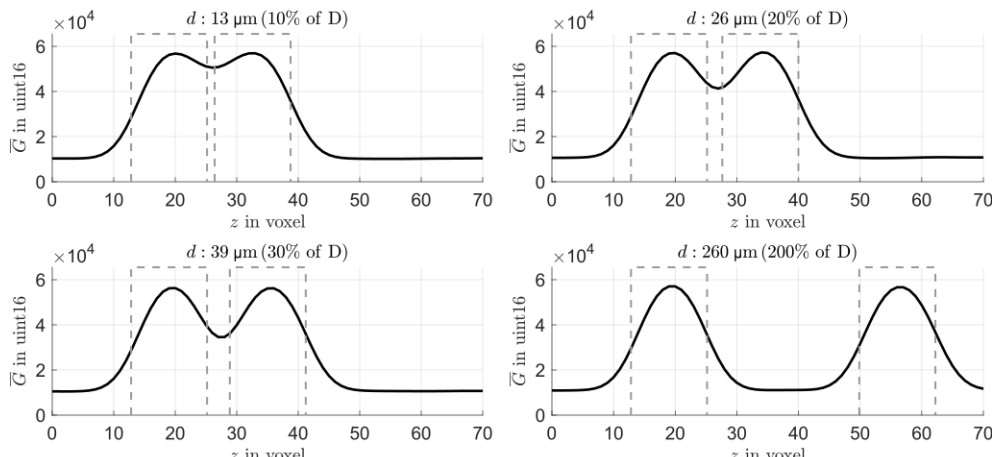

**Figure 6.** aRTist simulation of two book pages (cellulose, density: 1.5 g/cm$^3$) with a constant thickness of $D = 130$ μm indicated by the dashed lines and increasing distance d. The contrast signal $\overline{G}$ was averaged in the central area of the papers along $z$. The simulation was parameterized with the nominal values from Table 1 but with a smaller detector pixel pitch of 50 μm, resulting in a smaller voxel size of 10.5 μm to highlight the signal transition.

In case of the paper stack CT measurements, additional data pre-processing and signal refinements had to be applied, which are summarized in Figure 7.

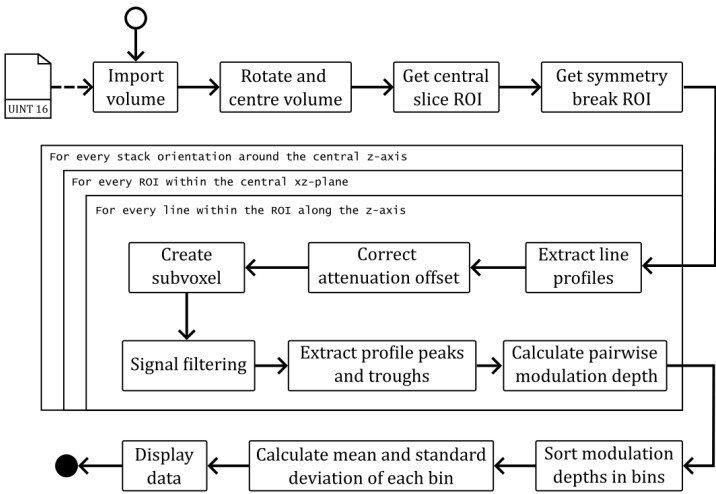

**Figure 7.** Evaluation process to calculate the modulation depth of paper stack CT measurements.

First, the volume is rotated in order to follow the global coordinate system convention of Figure 5. The punched spacers provide a good indicator in the $xy$-plane to define a circle and align the whole volume around its centre. After that, the central slice of the volume ($xz$-plane) is used to define a region of interest (ROI), which starts and ends within the starting/closing pattern and avoids edges of the punched spacers (compare Figure 8). In the same volume slice, an ROI of the symmetry break is defined, which will be used to process the signal.

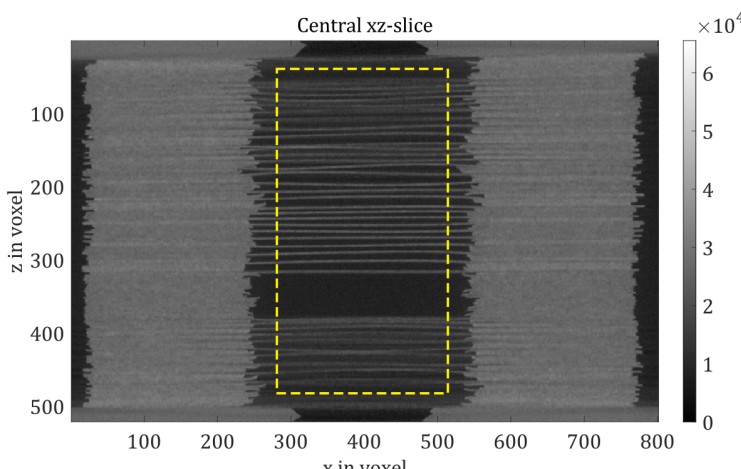

**Figure 8.** Midplane of the reconstructed volume (voxel size of 42.1 μm). Each line along *z* that is within the yellow ROI has been considered for evaluation.

After those pre-processing steps, the stack is rotated around the centred *z*-axis and each line along *z* of the defined ROI is evaluated separately. For each voxel on a line, the mean of the extracted symmetry break ROI is subtracted as an attenuation offset, cutting off some of the noise in between the gaps.

Due to remaining waviness of the stacked papers and the limited resolution of the voxel size, the voxel size had to be further reduced. Thus, each voxel was separated into three equal subvoxels with a third of the original size. In the next step, a moving mean along the profile line is applied with a cubic filter of $3 \times 3 \times 3$ subvoxel centred around the current subvoxel. The filter was chosen to be the same size as the subvoxel division to reduce the influence of interpolation.

From the filtered signal, the highest 38 peaks (36 from the variable patterns, 1 from the symmetry break, and 1 from the closing pattern) are selected, which do not overlap each other within $2 \cdot \frac{D}{2} = 130$ μm. The valleys are set to be the smallest grey value in between the selected peaks. Each peak/valley combination is used pairwise to calculate the peak-to-peak distance $d_{\text{P2P}}$ and the modulation depth $M_{\text{DIP}}$ according to Equation (1).

The complete set of $\{d_{\text{P2P}}; M_{\text{DIP}}\}$ for every line within the ROI for every rotation and every repeated measurement is finally sorted in $d_{\text{P2P}}$ bins with a width of 14 μm (size of the subvoxel). Finally, the mean and the standard deviation of each binned modulation depth are calculated (see Figure 9).

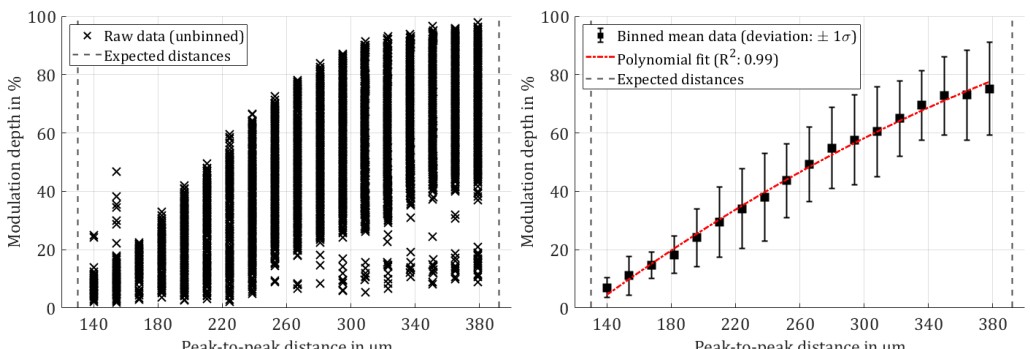

**Figure 9.** (**Left**) Resulting modulation depths of three repeated CT measurements. (**Right**) Binning of the raw data and calculation of the mean modulation depth, which can be fitted in good agreement with a second-order polynomial fit. Both: around 41% of the $9.44 \cdot 10^6$ samples are within the constructed peak-to-peak distance interval of $[130; 392]$ μm.

## 4. Results

The results of three repeated CT measurements of the paper stack, following the evaluation process described in Section 3.3, are summarized in Figure 9. A total of $9.44 \times 10^6$ peak-to-peak samples were analysed and sorted into 18 peak-to-peak bins within the constructed intervals of [130; 392] μm (Figure 9, right).

A polynomial fit (second-order) was fitted to the binned modulation depths (Figure 9, right), resulting in a high agreement ($R^2 = 0.99$) with the ASTM E2002-22 standard [17]. The progression of the fit curve shows a positive correlation between the mean modulation depth $M_{\text{DIP}}$ and the peak-to-peak distances $d_{\text{P2P}}$ of adjacent pages within a measured CT volume. However, the uniformly distributed large standard deviations indicate that further refinement of the evaluation process is required for a stable criterion, like a system-specific threshold, to assess the separability of arbitrarily oriented book pages.

## 5. Discussion

The presented method to assess the separability of arbitrary close pages provides several points for discussion. First, the successful application of the proposed methodology highlights its potential as a practical tool to evaluate the CT setup for a unique measurement case. This was demonstrated by the positive correlation between the modulation depth and the mean peak-to-peak distances.

Figure 9 illustrates the correlation between the peak-to-peak distances designed in the test object and the modulation depths obtained by evaluating the CT measurements. The majority of the modulation depths can be attributed to distances outside the designed distances. As previously described, these variations can be attributed to the inherent deformation of the paper pages due to the natural fibre structure. Although the page spacings can vary due to the natural deformation of the paper, a nearly linear behaviour can still be observed over the mean values of the spacings. This observation shows that, as the distance between the pages increases, the modulation depths also increase, confirming the simulated convergence behaviour (Figure 6). This result confirms the assumption that the correlation between page spacing and modulation depth follows a consistent pattern and can be utilized to define a fixed seperation criterion.

However, the presented study has some limitations. First, the analysis focused on the modulation depths associated with increasing page spacing. Other factors that could influence the observed behaviour, such as variations in paper thickness and surface roughness, were not considered. Future research could include these factors for a more complete understanding. In addition, it should be noted that commercially available paper grades were used for our test object to ensure reproducibility. The generalisability of our results to other paper grades might therefore be limited. It would be beneficial to repeat our experiments with a wider range of paper materials and sizes using different CT scanners to confirm the observed trends and determine if similar behaviour occurs.

It should also be noted that our study focused exclusively on the analysis of blank pages of commercial paper. The applicability of our results to other materials or substrates, as well as the influence of metallic inks, may require further investigation. This is because metallic inks may cause CT artefacts or interact with the paper through oxidation processes or chemical reactions, which may further complicate separation. Further investigation of the specimen, including the influence of historical inks, could be a valuable contribution to the further evaluation of the digital separation of book pages.

Finally, it is important to note that historical books exist in a variety of forms and types, each with its own unique characteristics. Furthermore, the condition of these books can vary significantly. While some historical documents are well preserved, others exhibit varying degrees of deterioration due to varying humidity, mould, insect damage, ink erosion, and other influences. Accordingly, it is essential to conduct comprehensive studies on authentic historical documents in order to verify the presented methodology and assess potential limitations in its applicability.

Despite these limitations, our study contributes to the existing body of knowledge by confirming the analysis of the modulation depth of a suitable test object as a possible separation criterion for historical documents. Considering these limitations in future research may help to refine the understanding gained and broaden the areas of application of these results.

## 6. Conclusions

In this study, we have introduced a novel approach for evaluating the suitability of an existing CT measurement system in the digitization of historical documents. Our primary objective was to develop a comprehensive and reliable method for estimating the spatial separation of book pages from a reconstructed volume of a scanned test object.

For this purpose, we designed a test object consisting of circular sheets of paper separated at different distances by paper rings of different thickness. This test object served two main purposes. Firstly, it can be used to determine the optimal scanning parameters in advance, thus minimising the radiation exposure to sensitive books during the actual scanning process. Secondly, it can be used to evaluate the suitability of the measurement system for accurate segmentation of book pages.

The test object was repeatedly scanned using CT and the volume data obtained were evaluated according to the methodology described. The aim was to integrate the modulation depth as an approach for separating scanned book pages and to test its suitability. For this purpose, 9.44 million inter-page distances were analysed and compared with the corresponding modulation depths. The result shows a clear correlation between the page distances and the corresponding modulation depth, which indicates that this is a suitable approach to be able to infer the separability of historical documents in advance. This finding is important because, without a digital separation of the scanned pages, it is difficult to extract the text on the pages later.

Overall, this study not only developed a new methodology for estimating page separation but also presented a test procedure that makes it possible to check an existing CT system in advance for its suitability for scanning historical documents and to determine suitable scanning parameters. In this way, this study makes a contribution to reduce radiation exposure during CT scanning of vulnerable historical documents. Further research and development should focus on evaluating the individual scanning parameters, refining the methodology and extending its applicability to different types of genuine historical documents. In this way, we can bring the field forward to preserve and digitise this fragile cultural heritage for future generations.

The code developed for this study will be available for open access on GitHub for better replicability. This will allow researchers and interested parties to view and replicate the code to gain a better understanding of the methodology and further the advancement of knowledge in this research field [18].

**Author Contributions:** P.Z. and F.B. contributed to conceptualization, data curation, formal analysis, investigation, methodology, software, validation, visualization, and writing the original draft. T.H. provided resources and supervision. T.H. and P.Z. acquired funding and administered the project. F.B., P.Z., and T.H. contributed to review and editing. All authors have read and agreed to the published version of the manuscript.

**Funding:** This research was funded by the Deutsche Forschungsgemeinschaft (DFG, German Research Foundation)—433501541.

**Data Availability Statement:** The data presented in this study are available on request from the corresponding author.

**Conflicts of Interest:** The authors declare no conflict of interest.

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
