# Peer review of "Preserving Fragile History: Assessing the Feasibility of Segmenting Digitized Historical Documents with Modulation Depth Analysis"

_heritage, doi:10.3390/heritage6100343_

Round 1

Reviewer 1 Report

The article " Preserving Fragile History: Assessing the Feasibility of Segmenting Digitized Historical Documents with Modulation Depth Analysis" introduces a pioneering approach to assess the suitability of CT measurement systems for digitizing delicate historical documents. The authors devised a test object resembling a book, simplifying the replication process. This object served to determine optimal scanning parameters and to evaluate the system's effectiveness in segmenting book pages.

The article exhibits clarity, articulate writing, and introduces a commendable level of novelty in scientific research. I take this opportunity to extend my congratulations to the authors for their great work.

I have a few minor suggestions to enhance the article's readability:

In the abstract and throughout the paper, the authors employ various terms to refer to the test object. I propose standardizing the terminology to either "test object", "book mock-up", "book model", or "facsimile", as these terms are more appropriate in a scientific context, rather than using “book dummy”.

On line 32, I recommend replacing "isn't" with "is not" for a more formal tone.

Line 44: The use of "concerned" in this context puzzled me. It might be more effective to express that ionizing radiation is known to induce damage.

On line 49, I suggest referring to the CT scan as an analytical tool for the investigation and study of books and similar manuscripts, rather than a "preserving" technique.

Regarding the statement on lines 87-88 concerning the round shape chosen to mitigate the effects of varying transmission lengths, I would appreciate further clarification on how this parameter, applicable to flat sheets, translates practically to the irregular shapes of actual books.

Lines 125-129 introduce the concept of symmetry break, which could potentially confuse a non-expert reader. It would be beneficial if the authors could elucidate this concept in a more accessible manner.

On line 258, I propose asserting that the applicability of these results to genuine artefacts will necessitate further investigation, particularly considering the diverse shapes of actual books and the presence of inks, as indicated by the authors.

In summary, this article stands as a remarkable contribution to the field, and It was a pleasure to read it and review it.

The quality of English is satisfying, only a few minor edits are needed to increase readability 

Author Response

Please see the attachment with the answer and the revised manuscript.

Reviewer 2 Report

Reviewer Recommendation and Comments for Manuscript  ID: heritage-2614723 entitled Preserving Fragile History: Assessing the Feasibility of Segmenting Digitized Historical Documents with Modulation Depth Analysis  for Heritage.

The manuscript entitled Preserving Fragile History: Assessing the Feasibility of Segmenting Digitized Historical Documents with Modulation Depth Analysis by P. Zippert, F. Binder and T. Hausotte is a novel and good work and was written with a sound purpose, showing interesting results. Therefore, I recommend publishing the paper.

Author Response

(The authors gave the same response as above.)

Reviewer 3 Report

The article discusses an important issue regarding the preservation of historical documents and proposes the use of computed tomography (CT) as a non-destructive method to examine their contents. I believe that this article is suitable for publication as it presents a novel approach to assessing the separability of pages in historical documents, which is a valuable contribution to the field. However, it is crucial to validate these findings by examining actual historical samples. Historical documents and books have different conditions compared to laboratory samples, with various variables such as varying degrees of damage, humidity levels, fungal effects, insect damage, and the type and quantity of paper filler, all potentially influencing the results. Therefore, it is necessary to evaluate historical samples in order to confirm and validate these findings. If the authors are unable to conduct such evaluations, they should thoroughly address these challenges in the article.

Author Response

(The authors gave the same response as above.)
